# Revisiting adversarial training for the worst-performing class

**Thomas Pethick**                                                    *thomas.pethick@epfl.ch*
*École Polytechnique Fédérale de Lausanne (EPFL)*

**Grigorios G Chrysos**                                               *grigorios.chrysos@epfl.ch*
*École Polytechnique Fédérale de Lausanne (EPFL)*

**Volkan Cevher**                                                     *volkan.cevher@epfl.ch*
*École Polytechnique Fédérale de Lausanne (EPFL)*

**Reviewed on OpenReview:** *https://openreview.net/forum?id=wkecshlYxI*

## Abstract

Despite progress in adversarial training (AT), there is a substantial gap between the top-performing and worst-performing classes in many datasets. For example, on CIFAR10, the accuracies for the best and worst classes are 74% and 23%, respectively. We argue that this gap can be reduced by explicitly optimizing for the worst-performing class, resulting in a min-max-max optimization formulation. Our method, called class focused online learning (CFOL), includes high probability convergence guarantees for the worst class loss and can be easily integrated into existing training setups with minimal computational overhead. We demonstrate an improvement to 32% in the worst class accuracy on CIFAR10, and we observe consistent behavior across CIFAR100 and STL10. Our study highlights the importance of moving beyond average accuracy, which is particularly important in safety-critical applications.

## 1 Introduction

The susceptibility of neural networks to adversarial attacks (Goodfellow et al., 2014; Szegedy et al., 2013) has been a grave concern over the launch of such systems in real-world applications. Defense mechanisms that optimize the average performance have been proposed (Papernot et al., 2016; Raghunathan et al., 2018; Guo et al., 2017; Madry et al., 2017; Zhang et al., 2019). In response, even stronger attacks have been devised (Carlini & Wagner, 2016; Engstrom et al., 2018; Carlini, 2019).

In this work, we argue that the average performance is not the only criterion that is of interest for real-world applications. For classification, in particular, optimizing the average performance provides very poor guarantees for the "weakest" class. This is critical in scenarios where we require *any* class to perform well. It turns out that the worst performing class can indeed be much worse than the average in adversarial training. This difference is already present in clean training but we critically observe, that the gap between the average and the worst is greatly exacerbated in adversarial training. This gap can already be observed on CIFAR10 where the accuracy across classes is far from uniform with 51% average robust accuracy while the worst class is 23% (see Figure 1). The effect is even more prevalent when more classes are present as in CIFAR100 where we observe that the worst class has *zero* accuracy while the average accuracy is 28% (see Appendix C where we include other datasets). Despite the focus on adverarial training, we note that the same effect can be observed for robust evaluation after *clean* training (see Figure 4 §C).

This dramatic drop in accuracy for the weakest classes begs for different approaches than the classical empirical risk minimization (ERM), which focuses squarely on the average loss. We suggest a simple alternative, which we call *class focused online learning* (CFOL), that can be plugged into existing adversarial training procedures. Instead of minimizing the average performance over the dataset we sample from an adversarial distribution over classes that is learned jointly with the model parameters. In this way we aim

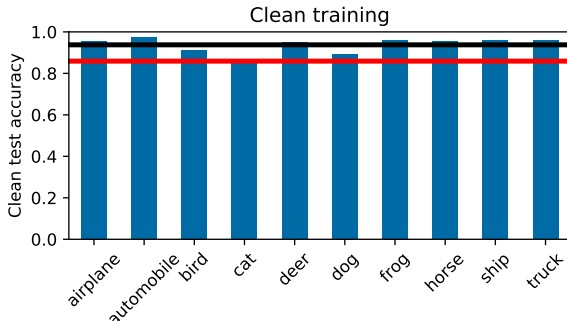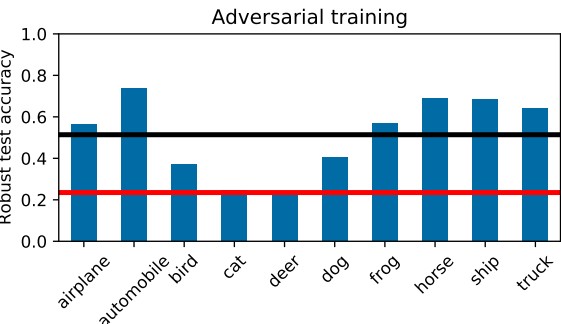

Figure 1: The error across classes is already not perfectly uniform in clean training on CIFAR10. However, this phenomenon is significantly worsened in adversarial training when considering the robust accuracy. That is, some classes perform much worse than the average. The worst class accuracy and average accuracy is depicted with a red and black line respectively.

at ensuring some level of robustness even for the worst performing class. The focus of this paper is thus on the robust accuracy of the weakest classes instead of the average robust accuracy.

Concretely, we make the following contributions:

- We propose a simple solution which relies on the classical bandit algorithm from the online learning literature, namely the Exponential-weight algorithm for Exploration and Exploitation (Exp3) (Auer et al., 2002). The method is directly compatible with standard adversarial training procedures (Madry et al., 2017), by replacing the empirical distribution with an adaptively learned adversarial distribution over classes.

- We carry out extensive experiments comparing CFOL against three strong baselines across three datasets, where we consistently observe that CFOL improves the weakest classes.

- We support the empirical results with high probability convergence guarantees for the worst class accuracy and establish direct connection with the conditional value at risk (CVaR) (Rockafellar et al., 2000) uncertainty set from distributional robust optimization.

## 2 Related work

**Adversarial examples** Goodfellow et al. (2014); Szegedy et al. (2013) are the first to make the important observation that deep neural networks are vulnerable to small adversarially perturbation of the input. Since then, there has been a growing body of literature addressing this safety critical issue, spanning from certified robust model (Raghunathan et al., 2018), distillation (Papernot et al., 2016), input augmentation (Guo et al., 2017), to adversarial training (Madry et al., 2017; Zhang et al., 2019). We focus on adversarial training in this paper. While certified robustness is desirable, adversarial training remains one of the most successful defenses in practice.

Parallel work (Tian et al., 2021) also observe the non-uniform accuracy over classes in adversarial training, further strengthening the case that lack of class-wise robustness is indeed an issue. They primarily focus on constructing an attack that can enlarge this disparity. However, they also demonstrate the possibility of a defense by showing that the accuracy of class 3 can be improved by manually reweighting class 5 when retraining. Our method CFOL can be seen as automating this process of finding a defense by instead adaptively assigning more weight to difficult classes.

**Minimizing the maximum** Focused online learning (FOL) (Shalev-Shwartz & Wexler, 2016), takes a bandit approach similar to our work, but instead re-weights the distribution over the $N$ training examples

independent of the class label. This leads to a convergence rate in terms of the number of examples $N$ instead of the number of classes $k$ for which usually $k \ll N$. We compare in more detail theoretically and empirically in Section 4.2 and Section 5 respectively. Sagawa et al. (2019) instead reweight the gradient over known groups which coincides with the variant considered in Section 4.1. A reweighting scheme has also been considered for data augmentation (Yi et al., 2021). They obtain a closed form solution under a heuristically driven entropy regularization and full information. In our setting full information would imply full batch updates and would be infeasible.

Interpolations between average and maximum loss have been considered in various other settings: for class imbalanced datasets (Lin et al., 2017), in federated learning (Li et al., 2019), and more generally the tilted empirical risk minimization (Li et al., 2020; Lee et al., 2020).

**Distributional robust optimization**  The accuracy over the worst class can be seen as a particular re-weighing of the data distribution which adversarially assigns all weights to a single class. Worst case perturbation of the data distribution have more generally been studied under the framework of distributional robust stochastic optimization (DRO) (Ben-Tal et al., 2013; Shapiro, 2017). Instead of attempting to minimizing the empirical risk on a training distribution $P_0$, this framework considers some *uncertainty set* around the training distribution $\mathcal{U}(P_0)$ and seeks to minimize the worst case risk within this set, $\sup_{Q \in \mathcal{U}(P_0)} \mathbb{E}_{x \sim Q}[\ell(x)]$.

A choice of uncertainty set, which has been given significant attention in the community, is conditional value at risk (CVaR), which aims at minimizing the weighted average of the tail risk (Rockafellar et al., 2000; Levy et al., 2020; Kawaguchi & Lu, 2020; Fan et al., 2017; Curi et al., 2019). CVaR has been specialized to a re-weighting over class labels, namely labeled conditional value at risk (LCVaR) (Xu et al., 2020). This was originally derived in the context of imbalanced dataset to re-balance the classes. It is still applicable in our setting and we thus provide a comparison. The original empirical work of (Xu et al., 2020) only considers the full-batch setting. We complement this by demonstrating LCVaR in a stochastic setting.

In Duchi et al. (2019); Duchi & Namkoong (2018) they are interested in uniform performance over various groups, which is similarly to our setting. However, these groups are assumed to be *latent* subpopulations, which introduce significant complications. The paper is thus concerned with a different setting, an example being training on a dataset implicitly consisting of multiple text corpora.

CFOL can also be formulated in the framework of DRO by choosing an uncertainty set that can re-weight the $k$ class-conditional risks. The precise definition is given in Appendix B.1. We further establish a direct connection between the uncertainty sets of CFOL and CVaR that we make precise in Appendix B.1, which also contains a summary of the most relevant related methods in Table 5 §B.

## 3  Problem formulation and preliminaries

**Notation**  The data distribution is denoted by $\mathcal{D}$ with examples $x \in \mathbb{R}^d$ and classes $y \in [k]$. A given iteration is characterized by $t \in [T]$, while $p_t^y$ indicates the $y^{\text{th}}$ index of the $t^{\text{th}}$ iterate. The indicator function is denoted with $\mathbb{1}_{\{\text{boolean}\}}$ and $\text{unif}(n)$ indicates the uniform distribution over $n$ elements. An overview of the notation is provided in Appendix A.

In classification, we are normally interested in minimizing the population risk $\mathbb{E}_{(x,y) \sim \mathcal{D}}[\ell(\theta, x, y)]$ over our model parameters $\theta \in \mathbb{R}^p$, where $\ell$ is some loss function of $\theta$ and example $x \in \mathbb{R}^d$ with an associated class $y \in [k]$. Madry et al. (2017) formalized the objective of adversarial training by replacing each example with an adversarially perturbed variant. That is, we want to find a parameterization $\theta$ of our predictive model which solves the following optimization problem:

$$\min_{\theta} L(\theta) := \mathbb{E}_{(x,y) \sim \mathcal{D}} \left[ \max_{\delta \in \mathcal{S}} \ell(\theta, x + \delta, y) \right], \tag{1}$$

where each $x$ is now perturbed by adversarial noise $\delta \in \mathcal{S} \subseteq \mathbb{R}^d$. Common choices of $\mathcal{S}$ include norm-ball constraints (Madry et al., 2017) or bounding some notion of perceptual distance (Laidlaw et al., 2020). When the distribution over classes is uniform this is implicitly minimizing the *average* loss over all class. This does

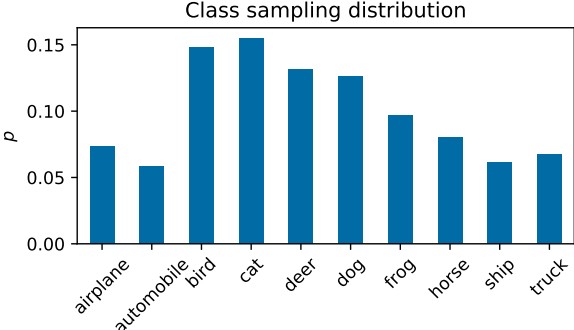

Figure 2: Contrary to ERM, which samples the examples uniformly, CFOL samples from an adaptive distribution. The learned adversarial distribution is non-uniform over the classes in CIFAR10 when using adversarial training. As expected, the hardest classes are also most frequently sampled.

not guarantee high accuracy for the *worst* class as illustrated in Figure 1, since we only know with certainty that $\max \geq \text{avg}$.

Instead, we will focus on a different objective, namely minimizing the *worst class-conditioned risk*:

$$\min_{\theta} \max_{y \in [k]} \left\{ L_y(\theta) := \mathbb{E}_{x \sim p_{\mathcal{D}}(\cdot|y)} \left[ \max_{\delta \in \mathcal{S}} \ell(\theta, x + \delta, y) \right] \right\}. \tag{2}$$

This follows the philosophy that "*a chain is only as strong as its weakest link*". In a safety critical application, such as autonomous driving, modeling even just a single class wrong can still have catastrophic consequences. Imagine for instance a street sign recognition system. Even if the system has 99% *average* accuracy the system might never label a "stop"-sign correctly, thus preventing the car from stopping at a crucial point.

As the maximum in Equation (2) is a discrete maximization problem its treatment requires more care. We will take a common approach and construct a convex relaxation by lifting the problem in Section 4.

## 4   Method

Since we do not have access to the true distribution $\mathcal{D}$, we will instead minimize over the provided empirical distribution. Let $\mathcal{N}_y$ be the set of data point indices for class $y$ such that the total number of examples is $N = \sum_{y=1}^{k} |\mathcal{N}_y|$. Then, we are interested in minimizing the *maximum empirical class-conditioned risk*,

$$\max_{y \in [k]} \widehat{L}_y(\theta) := \frac{1}{|\mathcal{N}_y|} \sum_{i \in \mathcal{N}_y} \max_{\delta \in \mathcal{S}} \ell(\theta, x_i + \delta, y). \tag{3}$$

We relax this discrete problem to a continuous problem over the simplex $\Delta_k$,

$$\max_{y \in [k]} \widehat{L}_y(\theta) \leq \max_{p \in \Delta_k} \sum_{y=1}^{k} p_y \widehat{L}_y(\theta). \tag{4}$$

Note that equality is attained when $p$ is a dirac on the argmax over classes.

Equation (4) leaves us with a two-player zero-sum game between the model parameters $\theta$ and the class distribution $p$. A principled way of solving a min-max formulation is through the use of no-regret algorithms. From the perspective of $p$, the objective is simply linear under simplex constraints, albeit adversarially picked

by the model. This immediately makes the no-regret algorithm Hedge applicable (Freund & Schapire, 1997):

$$w_y^t = w_y^{t-1} - \eta \widehat{L}_y(\theta^t),$$
$$p_y^t = \exp\left(w_y^t\right) / \sum_{y=1}^{k} \exp\left(w_y^t\right). \tag{Hedge}$$

To show convergence for (Hedge) the loss needs to satisfy certain assumptions. In our case of classification, the loss is the zero-one loss $\ell(\theta, x, y) = \mathbb{1}[h_\theta(x) \neq y]$, where $h_\theta(\cdot)$ is the predictive model. Hence, the loss is bounded, which is a sufficient requirement.

Note that (Hedge) relies on zero-order information of the loss, which we indeed have available. However, in the current form, (Hedge) requires so called *full information* over the $k$ dimensional loss vector. In other words, we need to compute $\widehat{L}_y(\theta)$ for all $y \in [k]$, which would require a full pass over the dataset for every iteration.

Following the seminal work of Auer et al. (2002), we instead construct an unbiased estimator of the $k$ dimensional loss vector $\widehat{L}(\theta^t) := (\widehat{L}_1(\theta^t), ..., \widehat{L}_k(\theta^t))^\top$ based on a sampled class $y^t$ from some distribution $y^t \sim p^t$. This stochastic formulation further lets us estimate the class conditioned risk $\widehat{L}_{y^t}(\theta^t)$ with an unbiased sample $i \sim \text{unif}(|\mathcal{N}_{y^t}|)$. This leaves us with the following estimator,

$$\widetilde{L}_y^t = \begin{cases} L_{y,i}(\theta^t)/p_y^t & y = y^t \\ 0 & \text{otherwise} \end{cases} \tag{5}$$

where $L_{y,i}(\theta) := \max_{\delta \in \mathcal{S}} \ell(\theta, x_i + \delta, y)$. It is easy to verify that this estimator is unbiased,

$$\mathbb{E}_{y \sim p}\left[\widetilde{L}_y^t\right] = \left(1 - p_y^t\right) \cdot 0 + p_y^t \cdot \frac{\widehat{L}_y(\theta^t)}{p_y^t} = \widehat{L}_y(\theta^t). \tag{6}$$

For ease of presentation the estimator only uses a single sample but this can trivially be extended to a mini-batch where classes are drawn i.i.d. from $p^t$.

We could pick $p^t$ to be the learned adversarial distribution, but it is well known that this can lead to unbounded regret if some $p_y^t$ is small (see Appendix A for definition of regret). Auer et al. (2002) resolve this problem with Exp3, which instead learns a distribution $q^t$, then mixes $q^t$ with a uniform distribution, to eventually sample from $p_y^t = \gamma \frac{1}{k} + (1-\gamma)q_y^t$ where $\gamma \in (0, 1)$. Intuitively, this enforces exploration. In the general case $\gamma$ needs to be picked carefully and small enough, but we show in Theorem 1 that a larger $\gamma$ is possible in our setting. We explore the effect of different choices of $\gamma$ empirically in Table 3.

Our algorithm thus updates $q^{t+1}$ with (Hedge) using the estimator $\widetilde{L}^t$ with a sample drawn from $p^t$ and subsequently computes $p^{t+1}$. CFOL in conjunction with the simultaneous update of the minimization player can be found in Algorithm 1 with an example of a learned distribution $p^t$ in Figure 2.

Practically, the scheme bears negligible computational overhead over ERM since the softmax required to sample is of the same dimensionality as the softmax used in the forward pass through the model. This computation is negligible in comparison with backpropagating through the entire model. For further details on the implementation we refer to Appendix C.3.

## 4.1 Reweighted variant of CFOL

Algorithm 1 samples from the adversarial distribution $p$. Alternatively one can sample data points uniformly and instead reweight the gradients for the model using $p$. In expectation, an update of these two schemes are equivalent. To see why, observe that in CFOL the model has access to the gradient $\nabla_\theta L_{y,i}(\theta)$. We can obtain an unbiased estimator by instead reweighting a uniformly sampled class, i.e. $\mathbb{E}_{y \sim p,i}\left[\nabla_\theta L_{y,i}(\theta)\right] = \mathbb{E}_{y \sim \text{unif}(k),i}\left[k p_y \nabla_\theta L_{y,i}(\theta)\right]$. With classes sampled uniformly the unbiased estimator for the adversary becomes $\widetilde{L}_{y'} = \mathbb{1}_{\{y'=y\}} L_y k \ \forall y'$. Thus, one update of CFOL and the reweighted variant are equivalent in expectation. However, note that we additionally depended on the internals of the model's update rule and that the

---

**Algorithm 1:** Class focused online learning (CFOL)

---

Algorithm parameters: a step rule ModelUpdate for the model satisfying Assumption 1, adversarial
  step-size $\eta > 0$, uniform mixing parameter $\gamma = 1/2$, and the loss $L_{y,i}(\theta)$.

Initialization: Set $w^0 = 0$ such that $q^0$ and $p^0$ are uniform.

**foreach** $t$ *in* $0..T$ **do**

$\quad\quad y^t \sim p^t$ ;                               `// sample class`

$\quad\quad i^t \sim \mathrm{unif}(|\mathcal{N}_{y^t}|)$ ;                `// sample uniformly from class`

$\quad\quad \theta^{t+1} = \mathrm{ModelUpdate}(\theta^t, L_{y^t,i^t}(\theta^t))$ ;       `// update model parameters`

$\quad\quad \widetilde{L}_y^t = \mathbb{1}_{\{y=y^t\}} L_{y,i^t}(\theta^t)/p_y^t \ \forall y$ ;           `// construct estimator`

$\quad\quad w_y^{t+1} = w_y^t - \eta \widetilde{L}_y^t \ \forall y$ ;       `// update the adv. class distribution`

$\quad\quad q_y^{t+1} = \exp\left(w_y^{t+1}\right)/\sum_{y=1}^k \exp\left(w_y^{t+1}\right) \ \forall y;$

$\quad\quad p_y^{t+1} = \gamma\frac{1}{k} + (1-\gamma)q_y^{t+1} \ \forall y;$

**end foreach**

---

immediate equivalence we get is only in expectation. These modifications recovers the update used in
Sagawa et al. (2019). See Table 9 in the supplementary material for experimental results regarding the
reweighted variant.

## 4.2 Convergence rate

To understand what kind of result we can expect, it is worth entertaining a hypothetical worst case scenario.
Imagine a classification problem where one class is much harder to model than the remaining classes. We
would expect the learning algorithm to require exposure to examples from the hard class in order to model
that class appropriately—otherwise the classes would not be distinct. From this one can see why ERM
might be slow. The algorithm would naively pass over the entire dataset in order to improve the hard class
using only the fraction of examples belonging to that class. In contrast, if we can adaptively focus on the
difficult class, we can avoid spending time on classes that are already improved sufficiently. As long as we
can adapt fast enough, as expressed through the regret of the adversary, we should be able to improve on
the convergence rate for the worst class.

We will now make this intuition precise by establishing a high probability convergence guarantee for the
worst class loss analogue to that of FOL. For this we will assume that the model parameterized by $\theta$ enjoys
a so called mistake bound of $C$ (Shalev-Shwartz et al., 2011, p. 288). The proof is deferred to Appendix A.

**Assumption 1.** For any sequence of classes $(y^1, ..., y^T) \in [k]^T$ and class conditioned sample indices $(i^1, ..., i^T)$
with $i^t \in \mathcal{N}_{y^t}$ the model enjoys the following bound for some $C' < \infty$ and $C = \max\{k \log k, C'\}$,

$$\sum_{t=1}^T L_{y^t,i^t}(\theta^t) \leq C. \tag{7}$$

*Remark* 1. The requirement $C \geq k \log k$ will be needed to satisfy the mild step-size requirement $\eta \leq 2k$ in
Lemma 1. In most settings the smallest $C'$ is some fraction of the number of iterations $T$, which in turn is
much larger than the number of classes $k$, so $C = C'$.

With this at hand we are ready to state the convergence of the worst class-conditioned empirical risk.

**Theorem 1.** *If Algorithm 1 is run on bounded rewards* $L_{y^t,i^t}(\theta^t) \in [0,1] \ \forall t$ *with step-size* $\eta = \sqrt{\log k/(4kC)}$,
*mixing parameter* $\gamma = 1/2$ *and the model satisfies Assumption 1, then after* $T$ *iterations with probability at
least* $1 - \delta$,

$$\max_{y \in [k]} \frac{1}{n} \sum_{j=1}^n \widehat{L}_y\left(\theta^{t_j}\right) \leq \frac{6C}{T} + \frac{\sqrt{4k \log(2k/\delta)}}{\sqrt{T}} + \frac{(1+2k)\log(2k/\delta)}{3T}$$

$$+ \frac{\sqrt{2\log(2k/\delta)}}{\sqrt{n}} + \frac{2\log(2k/\delta)}{3n}, \tag{8}$$

Table 1: Accuracy on CIFAR10. For both clean test accuracy ($\text{acc}_{\text{clean}}$) and robust test accuracy ($\text{acc}_{\text{rob}}$) we report the average, 20% worst classes and the worst class. We compare our method (CFOL-AT) with standard adversarial training (ERM-AT) and two baselines (LCVaR-AT and FOL-AT). CFOL-AT significantly improves the robust accuracy for both the worst class and the 20% tail.

|  |  | ERM-AT | CFOL-AT | LCVaR-AT | FOL-AT |
|---|---|---|---|---|---|
| $\text{acc}_{\text{clean}}$ | Average | 0.8244 | **0.8308** | 0.8259 | 0.8280 |
|  | 20% tail | 0.6590 | **0.7120** | 0.6540 | 0.6635 |
|  | Worst class | 0.6330 | **0.6830** | 0.6340 | 0.6210 |
| $\text{acc}_{\text{rob}}$ | Average | 0.5138 | 0.5014 | **0.5169** | 0.5106 |
|  | 20% tail | 0.2400 | **0.3300** | 0.2675 | 0.2480 |
|  | Worst class | 0.2350 | **0.3200** | 0.2440 | 0.2370 |

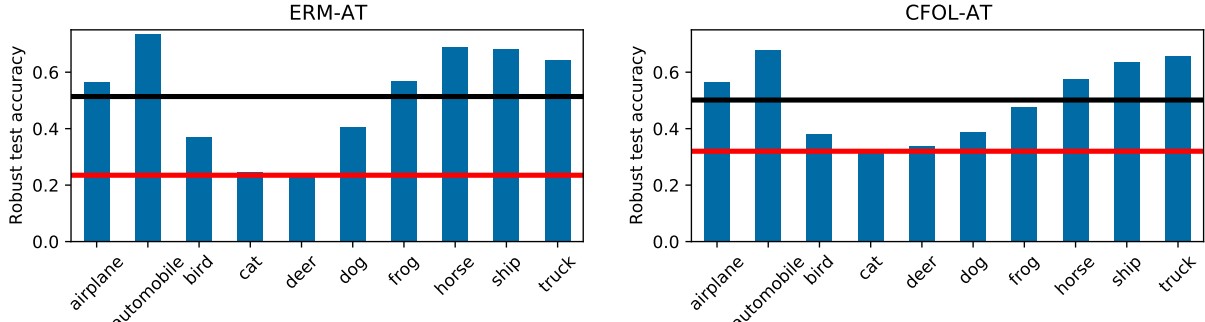

Figure 3: The robust test accuracy for CFOL-AT and ERM-AT over classes. The horizontal black and red line depicts the average and worst class accuracy over the classes respectively. The improvement in the minimum accuracy is notable when using CFOL-AT, while there is only marginal difference in the average accuracy. The evolution of accuracies for all methods can be found in Figure 7 §C.

*for an ensemble of size n where $t_j \overset{iid}{\sim} \text{unif}(T)$ for $j \in [n]$.*

To contextualize Theorem 1, let us consider the simple case of linear binary classification mentioned in Shalev-Shwartz & Wexler (2016). In this setting SGD needs $\mathcal{O}(CN)$ iterations to obtain a consistent hypothesis. In contrast, the iteration requirement of FOL decomposes into a sum $\widetilde{\mathcal{O}}(C + N)$ which is much smaller since both $C$ and $N$ are large. When we are only concerned with the convergence of the worst class we show that CFOL can converge as $\widetilde{\mathcal{O}}(C + k)$ where usually $k \ll C$. Connecting this back to our motivational example in the beginning of this section, this sum decomposition exactly captures our intuition. That is, adaptively focusing the class distribution can avoid the learning algorithm from needlessly going over all $k$ classes in order to improve just one of them.

From Theorem 1 we also see that we need an ensemble of size $n = \Omega(\sqrt{\log(k/\delta)}/\varepsilon^2)$, which has only mild dependency on the number of classes $k$. If we wanted to drive the worst class error $\varepsilon$ to zero the dependency on $\varepsilon$ would be problematic. However, for adversarial training, even in the best case of the CIFAR10 dataset, the *average* error is almost $1/2$. We can expect the worst class error to be even worse, so that only a small $n$ is required. In practice, a single model turns out to suffice.

Note that the maximum upper bounds the average, so by minimizing this upper bound as in Theorem 1, we are implicitly still minimizing the usual average loss. In addition, Theorem 1 shows that a mixing parameter of $\gamma = 1/2$ is sufficient for minimizing the worst class loss. Effectively, the average loss is still directly being minimized, but only through half of the sampled examples.

Table 2: Clean test accuracy ($\text{acc}_{\text{clean}}$) and robust test accuracy ($\text{acc}_{\text{rob}}$) on CIFAR100 and STL10. We compare our method (CFOL-AT) with standard adversarial training (ERM-AT) and two baselines (LCVaR-AT and FOL-AT). CFOL-AT consistently improves the worst class accuracy as well as the 20% worst tail.

| | | | **ERM-AT** | **CFOL-AT** | **LCVaR-AT** | **FOL-AT** |
|---|---|---|---|---|---|---|
| CIFAR100 | $\text{acc}_{\text{clean}}$ | Average | **0.5625** | 0.5593 | 0.5614 | 0.5605 |
| | | 20% tail | 0.2710 | **0.3550** | 0.2960 | 0.2815 |
| | | Worst class | 0.0600 | **0.2500** | 0.0600 | 0.1100 |
| | $\text{acc}_{\text{rob}}$ | Average | 0.2816 | 0.2519 | **0.2866** | 0.2718 |
| | | 20% tail | 0.0645 | **0.0770** | 0.0690 | 0.0630 |
| | | Worst class | 0.0000 | **0.0400** | 0.0000 | 0.0000 |
| STL10 | $\text{acc}_{\text{clean}}$ | Average | **0.7023** | 0.6826 | 0.6696 | 0.6890 |
| | | 20% tail | 0.4119 | **0.4594** | 0.3600 | 0.3837 |
| | | Worst class | 0.3725 | **0.4475** | 0.3462 | 0.3562 |
| | $\text{acc}_{\text{rob}}$ | Average | 0.3689 | 0.3755 | **0.3864** | 0.3736 |
| | | 20% tail | 0.0900 | **0.1388** | 0.0944 | 0.0981 |
| | | Worst class | 0.0587 | **0.1225** | 0.0650 | 0.0737 |

Table 3: Interpolation property of $\gamma$ on CIFAR10. By increasing the mixing parameter $\gamma$, CFOL-AT can closely match the average robust accuracy of ERM-AT while still improving the worst class accuracy. Standard deviations are computed over 3 independent runs.

| $\gamma$ | **0.9** | **0.7** | **0.5** |
|---|---|---|---|
| Average | **0.5108** $\pm$ 0.0015 | 0.5091 $\pm$ 0.0037 | 0.5012 $\pm$ 0.0008 |
| 20% tail | 0.2883 $\pm$ 0.0278 | 0.3079 $\pm$ 0.0215 | **0.3393** $\pm$ 0.0398 |
| Worst class | 0.2717 $\pm$ 0.0331 | 0.2965 $\pm$ 0.0141 | **0.3083** $\pm$ 0.0515 |

## 5 Experiments

We consider the adversarial setting where the constraint set of the attacker $\mathcal{S}$ is an $\ell_\infty$-bounded attack, which is the strongest norm-based attack and has a natural interpretation in the pixel domain. We test on three datasets with different dimensionality, number of examples per class and number of classes. Specifically, we consider CIFAR10, CIFAR100 and STL10 (Krizhevsky et al., 2009; Coates et al., 2011) (see Appendix C.2 for further details).

**Hyper-parameters** Unless otherwise noted, we use the standard adversarial training setup of a ResNet-18 network (He et al., 2016) with a learning rate $\tau = 0.1$, momentum of 0.9, weight decay of $5 \cdot 10^{-4}$, batch size of 128 with a piece-wise constant weight decay of 0.1 at epoch 100 and 150 for a total of 200 epochs according to Madry et al. (2017). For the attack we similarly adopt the common attack radius of 8/255 using 7 steps of projected gradient descent (PGD) with a step-size of 2/255 (Madry et al., 2017). For evaluation we use the stronger attack of 20 step throughout, except for Table 7 §C where we show robustness against AutoAttack (Croce & Hein, 2020).

**Baselines** With this setup we compare our proposed method, CFOL, against empirical risk minimization (ERM), labeled conditional value at risk (LCVaR) (Xu et al., 2020) and focused online learning (FOL) (Shalev-Shwartz & Wexler, 2016). We add the suffix "AT" to all methods to indicate that the training examples are adversarially perturbed according to adversarial training of Madry et al. (2017). We consider ERM-AT as the core baseline, while we also implement FOL-AT and LCVaR-AT as alternative methods that can improve the worst performing class. For fair comparison, and to match existing literature, we do early stopping based on the average robust accuracy on a hold-out set. The additional hyperparameters for CFOL-AT, FOL-AT and LCVaR-AT are chosen to optimize the robust accuracy for the worst class on CIFAR10. This parameter choice is then used as the basis for the subsequent datasets. More details on hyperparameters

Table 4: Comparison with larger pretrained models from the literature. ERM-AT refers to the training scheme of Madry et al. (2017) and uses the shared weights of the larger model from Engstrom et al. (2019). We refer to the pretrained Wide ResNet-34-10 from (Zhang et al., 2019) as *TRADES*. For fair comparison we apply CFOL-AT to a Wide ResNet-34-10 (with $\gamma = 0.8$), while using the test set as the hold-out set similarly to the pretrained models. We observe that CFOL-AT improves the worst class and 20% tail for both clean and robust accuracy.

|  |  | **TRADES** | **ERM-AT** | **CFOL-AT** |
|---|---|---|---|---|
| $\text{acc}_{\text{clean}}$ | Average | 0.8492 | 0.8703 | **0.8743** |
|  | 20% tail | 0.6845 | 0.7220 | **0.7595** |
|  | Worst class | 0.6700 | 0.6920 | **0.7500** |
| $\text{acc}_{\text{rob}}$ | Average | **0.5686** | 0.5490 | 0.5519 |
|  | 20% tail | 0.3445 | 0.3065 | **0.3575** |
|  | Worst class | 0.2810 | 0.2590 | **0.3280** |

and implementation can be found in Appendix C.1 and Appendix C.3 respectively. In Table 9 §C we additionally provide experiments for a variant of CFOL-AT which instead reweighs the gradients.

**Metrics** We report the average accuracy, the worst class accuracy and the accuracy across the 20% worst classes (referred to as the 20% tail) for both clean ($\text{acc}_{\text{clean}}$) and robust accuracy ($\text{acc}_{\text{rob}}$). We note that the aim is not to provide state-of-the-art accuracy but rather provide a fair comparison between the methods.

The first core experiment is conducted on CIFAR10. In Table 1 the quantitative results are reported with the accuracy per class illustrated in Figure 3. The results reveal that all methods other than ERM-AT improve the worst performing class with CFOL-AT obtaining higher accuracy in the weakest class than all methods in both the clean and the robust case. For the robust accuracy the 20% worst tail is improved from 24.0% (ERM-AT) to 26.7% using existing method in the literature (LCVaR-AT). CFOL-AT increases the former accuracy to 33.0% which is approximately a 40% increase over ERM-AT. The reason behind the improvement is apparent from Figure 3 which shows how the robust accuracy of the classes `cat` and `deer` are now comparable with `bird` and `dog`. A simple baseline which runs training twice is not able to improve the worst class robust accuracy (Table 11 §C).

The results for the remaining datasets can be found in Table 2. The results exhibit similar patterns to the experiment on CIFAR10, where CFOL-AT improves both the worst performing class and the 20% tail. For ERM-AT on STL10, the relative gap between the robust accuracy for the worst class (5.9%) and the average robust accuracy (36.9%) is even more pronounced. Interestingly, in this case CFOL-AT improves *both* the accuracy of the worst class to 12.3% and the average accuracy to 37.6%. In CIFAR100 we observe that both ERM-AT, LCVaR-AT and FOL-AT have 0% robust accuracy for the worst class, which CFOL-AT increases the accuracy to 4%.

In the case of CIFAR10, CFOL-AT leads to a minor drop in the average accuracy when compared against ERM-AT. For applications where average accuracy is also important, we note that the mixing parameter $\gamma$ in CFOL-AT interpolates between focusing on the average accuracy and the accuracy of the worst class. We illustrate this interpolation property on CIFAR10 in Table 3, where we observe that CFOL-AT can match the average accuracy of ERM-AT closely with 51% while still improving the worst class accuracy to 30%.

In the supplementary, we additionally assess the performance in the challenging case of a test time attack that differs from the attack used for training (see Table 7 §C for evaluation on AutoAttack). The difference between CFOL-AT and ERM-AT becomes even more pronounced. The worst class accuracy for ERM-AT drops to as low as 12.6% on CIFAR10, while CFOL-AT improves on this accuracy by more than 60% to an accuracy of 20.7%. We also carry out experiments under a larger attack radius (see Table 8 §C). Similarly, we observe an accuracy drop of the worst class on CIFAR10 to 9.3% for ERM-AT, which CFOL-AT improves to 17%. We also conduct experiments with the VGG-16 architecture, on Tiny ImageNet, Imagenette and under class imbalanced (Tables 12 to 15 §C). In all experiments CFOL-AT *consistently* improves the accuracy with respect to both the worst class and the 20% tail. We additionally provide results for early

stopped models using the best worst class accuracy from the validation set (see Table 6 §C). Using the worst class for early stopping improves the worst class accuracy as expected, but is not sufficient to make ERM-AT consistently competitive with CFOL-AT.

Even when compared with *larger* pretrained models from the literature we observe that CFOL-AT on the smaller ResNet-18 model has a better worst class robust accuracy (see Table 4 for the larger models). For fair comparison we also apply CFOL-AT to the larger Wide ResNet-34-10 model, while using the test set as the hold-out set similarly to the pretrained models. We observe that CFOL-AT improves both the worst class and 20% tail when compared with both pretrained ERM-AT and TRADES (Zhang et al., 2019). Whereas pretrained ERM-AT achieves 30.65% accuracy for the 20% tail, CFOL-AT achieves 35.75%. While the average robust accuracy of TRADES is higher, this is achieved by trading off clean accuracy.

**Discussion** We find that CFOL-AT consistently improves the accuracy for both the worst class and the 20% tail across the three datasets. When the mixing parameter $\gamma = 0.5$, this improvement usual comes at a cost of the average accuracy. However, we show that CFOL-AT can closely match the average accuracy of ERM-AT by picking $\gamma$ larger, while maintaining a significant improvement for the worst class.

Overall, our experimental results validate our intuitions that (a) the worst class performance is often overlooked and this might be a substantial flaw for safety-critical applications and (b) that algorithmically focusing on the worst class can alleviate this weakness. Concretely, CFOL provides a simple method for improving the worst class, which can easily be added on top of existing training setups.

## 6 Conclusion

In this work, we have introduced a method for class focused online learning (CFOL), which samples from an adversarially learned distribution over classes. We establish high probability convergence guarantees for the worst class using a specialized regret analysis. In the context of adversarial examples, we consider an adversarial threat model in which the attacker can choose the class to evaluate in addition to the perturbation (as shown in Equation (2)). This formulation allows us to develop a training algorithm which focuses on being robust across *all* the classes as oppose to only providing robustness on average. We conduct a thorough empirical validation on five datasets, and the results consistently show improvements in adversarial training over the worst performing classes.

## 7 Acknowledgements

This project has received funding from the European Research Council (ERC) under the European Union's Horizon 2020 research and innovation programme (grant agreement n° 725594 - time-data). This work was supported by Zeiss. This work was supported by Hasler Foundation Program: Hasler Responsible AI (project number 21043).

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

# A  Convergence analysis

## A.1  Preliminary and notation

Consider the abstract online learning problem where for every $t \in [T]$ the player chooses an action $x^t$ and subsequently the environment reveals the loss function $l(\cdot, y^t)$. As traditional in the online learning literature we will measure performance in terms of regret, which compares our sequence of choices $\{x^t\}_{t=1}^T$ with a fixed strategy in hindsight $u$,

$$\mathcal{R}_T(u) = \sum_{t=1}^T l(x^t, y^t) - \min_x \sum_{t=1}^T l(u, y^t). \tag{9}$$

If we, instead of minimizing over losses $l$, maximize over rewards $r$ we define regret as,

$$\mathcal{R}_T(u) = \max_x \sum_{t=1}^T r(u, y^t) - \sum_{t=1}^T r(x^t, y^t). \tag{10}$$

When we allow randomized strategy, as in the bandit setting, $\mathcal{R}_T$ becomes a random variable that we wish to upper bound with high probability. For convenience we include an overview over the notation defined and used in Section 3 and Section 4 below.

| | |
|---|---|
| $k$ | Number of classes |
| $T$ | Number of iterations |
| $p_y^t$ | The $y^{\text{th}}$ index of the $t^{\text{th}}$ iterate |
| $\mathbb{1}_{\{\text{boolean}\}}$ | The indicator function |
| $\text{unif}(n)$ | Discrete uniform distribution over $n$ elements |
| $\mathcal{N}_y$ | Set of data point indices for class $y \in [k]$ |
| $N = \sum_{y=1}^k |\mathcal{N}_y|$ | The size of the data set |
| $L_{y,i}(\theta) := \max_{\delta \in \mathcal{S}} \ell(\theta, x_i + \delta, y)$ | Loss on a particular example |
| $\widehat{L}_y(\theta) := \frac{1}{|\mathcal{N}_y|} \sum_{i \in \mathcal{N}_y} L_{y,i}(\theta)$ | The empirical class-conditioned risk |
| $\widehat{L}(\theta) := (\widehat{L}_1(\theta), \ldots, \widehat{L}_k(\theta))^\top$ | The vector of all empirical class-conditioned risks |
| $L_y^t := L_{y,i}(\theta^t)$ | Class-conditioned estimator at iteration $t$ with $i \sim \text{unif}(|\mathcal{N}_y|)$ |

## A.2  Convergence results

We restate Algorithm 1 while leaving out the details of the classifier for convenience. Initialize $w^0$ such that $q^0$ and $p^0$ are uniform. Then Exp3 proceeds for every $t \in [T]$ as follows:

1. Draw class $y^t \sim p^t$

2. Observe a scalar reward $L_{y^t}^t$

3. Construct estimator $\widetilde{L}_y^t = L_{y^t}^t \mathbb{1}_{\{y=y^t\}} / p_y^t \ \forall y$

4. Update distribution
   $w^{t+1} = w^t - \eta \widetilde{L}^t$
   $q_y^{t+1} = \exp\left(w_y^{t+1}\right) / \sum_{y=1}^k \exp\left(w_y^{t+1}\right) \ \forall y$
   $p_y^{t+1} = \gamma \frac{1}{k} + (1 - \gamma) q_y^{t+1} \ \forall y$

We can bound the regret of Exp3 (Auer et al., 2002) in our setting, even when the mixing parameter $\gamma$ is not small as otherwise usually required, by following a similar argument as Shalev-Shwartz & Wexler (2016).

For this we will use the relationship between $p$ and $q$ throughout. From $p_y = \frac{\gamma}{k} + (1-\gamma)q_y$ it can easily be verified that,

$$\frac{1}{p_y} \leq \frac{k}{\gamma} \text{ and } \frac{q_y}{p_y} \leq \frac{1}{1-\gamma} \text{ for all } y. \tag{11}$$

**Lemma 1.** *If Exp3 is run on bounded rewards $L_{y^t}^t \in [0,1]$ $\forall t$ with $\eta \leq \gamma/k$ then*

$$\mathcal{R}_T^{\text{adv}}(u) = \sum_{t=1}^{T} \langle u, \widetilde{L}^t \rangle - \sum_{t=1}^{T} \langle q^t, \widetilde{L}^t \rangle \leq \frac{\log k}{\eta} + \frac{\eta k}{(1-\gamma)\gamma} \sum_{t=1}^{T} L_y^t \tag{12}$$

*Proof.* We can write the estimator vector as $\widetilde{L}^t = \frac{L_{y^t}^t}{p_{y^t}^t} e_{y^t}$ where $e_{y^t}$ is a canonical basis vector. By assumption we have $L_{y^t}^t \in [0,1]$. Combining this with the bound on $1/p_y$ in Equation (11) we have $\widetilde{L}_y^t \leq k/\gamma$ for all $y$. This lets us invoke (Shalev-Shwartz et al., 2011, Thm 2.22) as long as the step-size $\eta$ is small enough. Specifically, we have that if $\eta \leq 1/\widetilde{L}_y^t \leq \gamma/k$

$$\sum_{t=1}^{T} \langle u, \widetilde{L}^t \rangle - \sum_{t=1}^{T} \langle q^t, \widetilde{L}^t \rangle \leq \frac{\log k}{\eta} + \eta \sum_{t=1}^{T} \sum_{y=1}^{k} q_y^t (\widetilde{L}_y^t)^2. \tag{13}$$

By expanding $\widetilde{L}_y^t$ and applying the inequalities from Equation (11) we can get rid of the dependency on $q$ and $p$,

$$
\begin{aligned}
\frac{\log k}{\eta} + \eta \sum_{t=1}^{T} \sum_{y=1}^{k} q_y^t (\widetilde{L}_y^t)^2 &\leq \frac{\log k}{\eta} + \eta \sum_{t=1}^{T} \frac{q_{y^t}^t}{(p_{y^t}^t)^2} (L_{y^t}^t)^2 \\
&\leq \frac{\log k}{\eta} + \eta \sum_{t=1}^{T} \frac{k}{(1-\gamma)\gamma} (L_{y^t}^t)^2 \\
&\leq \frac{\log k}{\eta} + \frac{\eta k}{(1-\gamma)\gamma} \sum_{t=1}^{T} L_{y^t}^t,
\end{aligned}
\tag{14}
$$

where the last line uses the boundedness assumption $L_y^t \in [0,1]$ $\forall y$. $\qquad\square$

It is apparent that we need to control the sum of losses. We will do so by assuming that the model admits a mistake bound as in Shalev-Shwartz & Wexler (2016); Shalev-Shwartz et al. (2011).

**Assumption 1.** *For any sequence of classes $(y^1, ..., y^T) \in [k]^T$ and class conditioned sample indices $(i^1, ..., i^T)$ with $i^t \in \mathcal{N}_{y^t}$ the model enjoys the following bound for some $C' < \infty$ and $C = \max\{k \log k, C'\}$,*

$$\sum_{t=1}^{T} L_{y^t, i^t}(\theta^t) \leq C. \tag{15}$$

Recall that we need to bound $L_y^t := L_{y,i^t}(\theta^t)$, where $y$ are picked adversarially and $i^t$ are sampled uniformly, so the above is sufficient.

**Lemma 2.** *If the model satisfies Assumption 1 and Algorithm 1 is run on bounded rewards $L_{y^t, i^t}(\theta^t) \in [0,1]$ $\forall t$ with step-size $\eta = \sqrt{\log k/(4kC)}$ and mixing coefficient $\gamma = 1/2$ then*

$$\sum_{t=1}^{T} \langle u, \widetilde{L}^t \rangle \leq 6C. \tag{16}$$

*Proof.* By expanding and rearranging Lemma 1,

$$\sum_{t=1}^{T} \langle u, \widetilde{L}^t \rangle \leq \frac{\log k}{\eta} + \left( \frac{\eta k}{(1-\gamma)\gamma} + \frac{q_{y^t}^t}{p_{y^t}^t} \right) \sum_{t=1}^{T} L_y^t, \tag{17}$$

as long as $\eta \leq \gamma/k$. Then by using our favorite inequality in Equation (11) and under Assumption 1,

$$\frac{\log k}{\eta} + \left( \frac{\eta k}{(1-\gamma)\gamma} + \frac{q_{y^t}^t}{p_{y^t}^t} \right) \sum_{t=1}^{T} L_y^t \leq \frac{\log k}{\eta} + \left( \frac{\eta k}{(1-\gamma)\gamma} + \frac{1}{1-\gamma} \right) C, \tag{18}$$

To maximize $(1-\gamma)\gamma$ we now pick $\gamma = 1/2$ so that,

$$\frac{\log k}{\eta} + \left( \frac{\eta k}{(1-\gamma)\gamma} + \frac{1}{1-\gamma} \right) C \leq \frac{\log k}{\eta} + (4\eta k + 2) C. \tag{19}$$

For $\eta$, notice that it appears in two of the terms. To minimize the bound respect to $\eta$ we pick $\eta = \sqrt{\log k/(4kC)}$ such that $\frac{\log k}{\eta} = 4\eta k C$, which leaves us with,

$$\frac{\log k}{\eta} + (4\eta k + 2)C \leq 2\sqrt{4kC \log k} + 2C. \tag{20}$$

From the first term it is clear that since $C \geq k \log k$ by assumption then the original step-size requirement of $\eta \leq 1/2k$ is satisfies. In this case we can additionally simplify further,

$$2\sqrt{4kC \log k} + 2C \leq 6C. \tag{21}$$

$\square$

This still only gives us a bound on a stochastic object. We will now relate it to the empirical class conditional risk $\widehat{L}_y(\theta)$ by using standard concentration bounds. To be more precise, we want to show that by picking $u = e_y$ in $\langle u, \widetilde{L}^t \rangle$ we concentrate to $\widehat{L}_y(\theta)$. Following Shalev-Shwartz & Wexler (2016) we adopt their use of a Bernstein's type inequality.

**Lemma 3** (e.g. Audibert et al. (2010, Thm. 1.2)). *Let $A_1, ..., A_T$ be a martingale difference sequence with respect to a Markovian sequence $B_1, ..., B_T$ and assume $|A_t| \leq V$ and $\mathbb{E}\left[ A_t^2 \mid B_1, \ldots, B_t \right] \leq s$. Then for any $\delta \in (0, 1)$,*

$$\mathbb{P}\left( \frac{1}{T} \sum_{t=1}^{T} A_t \leq \frac{\sqrt{2s \log(1/\delta)}}{\sqrt{T}} + \frac{V \log(1/\delta)}{3T} \right) \geq 1 - \delta. \tag{22}$$

**Lemma 4.** *If Algorithm 1 is run on bounded rewards $L_{y^t, i^t}(\theta^t) \in [0, 1]$ $\forall t$ with step-size $\eta = \sqrt{\log k/(4kC)}$, mixing coefficient $\gamma = 1/2$ and the model satisfies Assumption 1, then for any $y \in [k]$ we obtain have the following bound with probability at least $1 - \delta/k$,*

$$\frac{1}{T} \sum_{t=1}^{T} \widehat{L}_y\left( \theta^t \right) \leq \frac{6C}{T} + \frac{\sqrt{4k \log(k/\delta)}}{\sqrt{T}} + \frac{(1+2k)\log(k/\delta)}{3T}. \tag{23}$$

*Proof.* Pick any $y \in [k]$ and let $u = e_y$ in $\langle u, \widetilde{L}^t \rangle$. By construction the following defines a martingale difference sequence,

$$A_t = \widehat{L}_y(\theta^t) - \langle e_y, \widetilde{L}^t \rangle = \widehat{L}_y(\theta^t) - \langle e_y, \frac{1}{p_{y^t}^t} L_{y^t, i^t}(\theta^t) e_{y^t} \rangle. \tag{24}$$

In particular note that $i^t$ is uniformly sampled. To apply the Bernstein's type inequality we just need to bound $|A_t|$ and $\mathbb{E}\left[ A_t^2 \mid q^t, \theta^t \right]$. For $|A_t|$ we can crudely bound it as,

$$|A_t| \leq |\widehat{L}_y(\theta^t)| + |\langle e_y, \frac{1}{p_{y^t}^t} L_{y^t, i^t}(\theta^t) e_{y^t} \rangle| \leq 1 + \frac{k}{\gamma}. \tag{25}$$

To bound the variance observe that we have the following:

$$\mathbb{E}\left[\langle e_y, \widetilde{L}^t\rangle^2 \mid q^t, \theta^t\right] \le \sum_{y'=1}^{k} \frac{p_{y'}^t}{(p_{y'}^t)^2}\mathbb{E}\left[L_{y',i}(\theta^t)^2 \mid \theta^t\right](e_y)_{y'} \qquad \text{(with } i \sim \mathrm{unif}(|\mathcal{N}_{y'}|) \; \forall y')$$

$$= \frac{1}{p_y^t}\mathbb{E}\left[L_{y,i}\left(\theta^t\right)^2 \mid \theta^t\right]$$

$$\le \frac{1}{p_y^t} \le \frac{k}{\gamma}. \qquad \text{(by Equation (11))}$$

It follows that $\mathbb{E}\left[A_t^2 \mid q^t, \theta^t\right] \le k/\gamma$

Invoking Lemma 3 we get the following bound with probability at least $1 - \delta/k$,

$$\frac{1}{T}\sum_{t=1}^{T}\widehat{L}_y\left(\theta^t\right) \le \frac{1}{T}\sum_{t=1}^{T}\langle e_y, \widetilde{L}^t\rangle + \frac{\sqrt{2\frac{k}{\gamma}\log\left(k/\delta\right)}}{\sqrt{T}} + \frac{(1+\frac{k}{\gamma})\log\left(k/\delta\right)}{3T} \tag{26}$$

By bounding the first term on the right hand side with Lemma 2 and taking $\gamma = 1/2$ we obtain,

$$\frac{1}{T}\sum_{t=1}^{T}\widehat{L}_y\left(\theta^t\right) \le \frac{6C}{T} + \frac{\sqrt{4k\log\left(k/\delta\right)}}{\sqrt{T}} + \frac{(1+2k)\log\left(k/\delta\right)}{3T} \tag{27}$$

This completes the proof. $\qquad\square$

We are now ready to state the main theorem.

**Theorem 1.** *If Algorithm 1 is run on bounded rewards $L_{y^t,i^t}(\theta^t) \in [0,1]$ $\forall t$ with step-size $\eta = \sqrt{\log k/(4kC)}$, mixing parameter $\gamma = 1/2$ and the model satisfies Assumption 1, then after $T$ iterations with probability at least $1 - \delta$,*

$$\max_{y \in [k]}\frac{1}{n}\sum_{j=1}^{n}\widehat{L}_y\left(\theta^{t_j}\right) \le \frac{6C}{T} + \frac{\sqrt{4k\log(2k/\delta)}}{\sqrt{T}} + \frac{(1+2k)\log(2k/\delta)}{3T} + \frac{\sqrt{2\log(2k/\delta)}}{\sqrt{n}} + \frac{2\log(2k/\delta)}{3n}, \tag{28}$$

*for an ensemble of size $n$ where $t_j \overset{iid}{\sim} \mathrm{unif}(T)$ for $j \in [n]$.*

*Proof.* If we fix $y$ and let $t_j \overset{iid}{\sim} \mathrm{unif}(T)$, then the following is a martingale difference sequence,

$$A_j = \widehat{L}_y\left(\theta^{t_j}\right) - \frac{1}{T}\sum_{t=1}^{T}\widehat{L}_y\left(\theta^t\right), \tag{29}$$

for which it is easy to see that $|A_j| \le 2$ and $A_j^2 \le 1$ given boundedness of the loss. This readily let us apply Lemma 3 with high probability $1 - \delta/2k$. Combining this with the bound of Lemma 4 with probability $1 - \delta/2k$ by using a union bound completes the proof. $\qquad\square$

# B   CVaR

Conditional value at risk (CVaR) is the expected loss conditioned on being larger than the $(1-\alpha)$-quantile. This has a distributional robust interpretation which for a discrete distribution can be written as,

$$\text{CVaR}_\alpha\left(\theta, P_0\right) := \sup_{p \in \Delta_m} \left\{ \sum_{i=1}^m p_i \ell\left(\theta, x_i\right) \text{ s.t. } \|p\|_\infty \leq \frac{1}{\alpha m} \right\}. \tag{30}$$

The optimal $p$ of the above problem places uniform mass on the tail. Practically we can compute this best response by sorting the losses $\{\ell(\theta, x_i)\}_i$ in descending order and assigning $\frac{1}{\alpha m}$ mass per index until saturation.

**Primal CVaR**   When $m$ is large a stochastic variant is necessary. To obtain a stochastic subgradient for the model parameter, the naive approach is to compute a stochastic best response over a mini-batch. This has been studied in detail in (Levy et al., 2020).

**Dual CVaR**   An alternative formulation relies on strong duality of CVaR originally showed in (Rockafellar et al., 2000),

$$\text{CVaR}_\alpha\left(\theta, P_0\right) = \inf_{\lambda \in \mathbb{R}} \left\{ \lambda + \frac{1}{\alpha m} \sum_{i=1}^m \left(\ell\left(\theta, x_i\right) - \lambda\right)_+ \right\}. \tag{31}$$

In practice, one approach is to jointly minimize $\lambda$ and the model parameters by computing stochastic gradients as done in (Curi et al., 2019). Alternatively, we can find a close form solution for $\lambda$ under a mini-batch (see e.g. (Xu et al., 2020, Appendix I.1)).

In comparison CFOL acts directly on the probability distribution similarly to primal CVaR. However, instead of finding a best response on a uniformly sampled mini-batch it updates the weights iteratively and samples accordingly (see Algorithm 1). Despite this difference, it is interesting that a direct connection can be established between the uncertainty sets of the methods. The following section is dedicated to this.

To be pedantic it is worth pointing out a minor discrepancy when applying CVaR in the context of deep learning. Common architectures such as ResNets incorporates batch normalization which updates the running mean and variance on the forward pass. The implication is that the entire mini-batch is used to update these statistics despite the gradient computation only relying on the worst subset. This makes implementation in this setting slightly more convoluted.

## B.1   Relationship with CVaR

Exp3 is run with a uniform mixing to enforce exploration. This turns out to imply the necessary and sufficient condition for CVaR. We make this precise in the following lemma:

**Lemma 5.** *Let $p$ be a uniform mixing $p = \gamma \frac{1}{m} + (1-\gamma)q$ with an arbitrary distribution $q \in \Delta_m$ and $\epsilon \in [0,1]$ such that the distribution has a lower bound on each element $p_i \geq \gamma \frac{1}{m}$. Then an upper bound is implicitly implied on each element, $\|p\|_\infty := \max_{i=[m]} p_i \leq 1 - \frac{(m-1)\gamma}{m}$.*

*Proof.* Consider $p_i$ for any $i$. At least $(m-1)\frac{\gamma}{m}$ of the mass must be on other components so $p_i \leq 1 - \frac{(m-1)\gamma}{m}$. The result follows. □

**Corollary 1.** *Consider the uncertainty set of Exp3,*

$$\mathcal{U}^{\text{Exp3}}\left(P_0\right) = \left\{ p \in \Delta_m \mid p_i \geq \gamma \frac{1}{m} \ \forall i \right\}. \tag{32}$$

*Given that the above lower bound is only introduced for practical reasons, we might as well consider an instantiation of CVaR which turns out to be a proper relaxation,*

$$\mathcal{U}^{\text{CVaR}}\left(P_0\right) = \left\{ p \in \Delta_m \mid \|p\|_\infty \leq \frac{1}{\alpha m} \right\} \tag{33}$$

Table 5: Summary of methods where $N$ is the number of samples and $k$ is the number of classes. The discrete distribution $p$ either governs the distribution over all $N$ samples or over the $k$ classes depending on the algorithm (see columns).

| Uncertainty set | Over data point ($m = N$) | Over class labels ($m = k$) |
|---|---|---|
| $\mathcal{U}^{\mathrm{Exp3}}(P_0) = \left\{ p \in \Delta_m \mid p_i \geq \varepsilon \frac{1}{m} \ \forall i \right\}$ | FOL (Shalev-Shwartz & Wexler, 2016) | CFOL (ours) |
| $\mathcal{U}^{\mathrm{CVaR}}(P_0) = \left\{ p \in \Delta_m \mid \|p\|_\infty \leq \frac{1}{\alpha m} \right\}$ | CVaR (Levy et al., 2020) | LCVaR (Xu et al., 2020) |

with $\alpha = \frac{1}{(1-\gamma)m+\gamma}$. *This leaves us with the following primal formulation,*

$$\mathrm{CVaR}_\alpha (\theta, P_0) := \sup_{p \in \Delta_m} \left\{ \sum_{i=1}^{m} p_i \ell(\theta, x_i) \ \ s.t. \ \|p\|_\infty \leq \frac{1}{\alpha m} \right\}. \tag{34}$$

*Proof.* From Lemma 5 and since CVaR requires $\|p\|_\infty \leq \frac{1}{\alpha m}$ we have,

$$\|p\|_\infty \leq 1 - \frac{(m-1)\gamma}{m} =: \frac{1}{\alpha m} \tag{35}$$

By simple algebra we have,

$$\alpha = \frac{1}{(1-\gamma)m + \gamma}, \tag{36}$$

which completes the proof. $\qquad\qquad\qquad\qquad\qquad\qquad\qquad\qquad\qquad\qquad\qquad\qquad\qquad\qquad\square$

So if the starting point of the uncertainty set is the simplex and the uniform mixing in Exp3 is therefore only for tractability reasons, then we might as well minimize the CVaR objective instead. This could even potentially lead to a more robust solution as the uncertainty set is larger (since the upper bound in $\mathcal{U}^{\mathrm{CVaR}}$ does not imply the lower bound in $\mathcal{U}^{\mathrm{Exp3}}$).

There are two things to keep in mind though. First, $\alpha$ should not be too small since the optimization problem gets harder. It is informative to consider the case where $\gamma = 1/2$ such that $\alpha = \frac{2}{m+1}$. From this it becomes clear that the recasting as CVaR only works for small $m$. Secondly, despite the uncertainty sets being related, the training dynamics, and thus the obtained solution, can be drastically different, as we also observe experimentally in Section 5. For instance CVaR is known for having high variance (Curi et al., 2019) while the uniform mixing in CFOL prevents this. It is worth noting that despite this uniform mixing we are still able to show convergence for CFOL in terms of the worst class in Section 4.2.

In Table 5 we provide an overview of the different methods induced by the choice of uncertainty set.

## C   Experiments

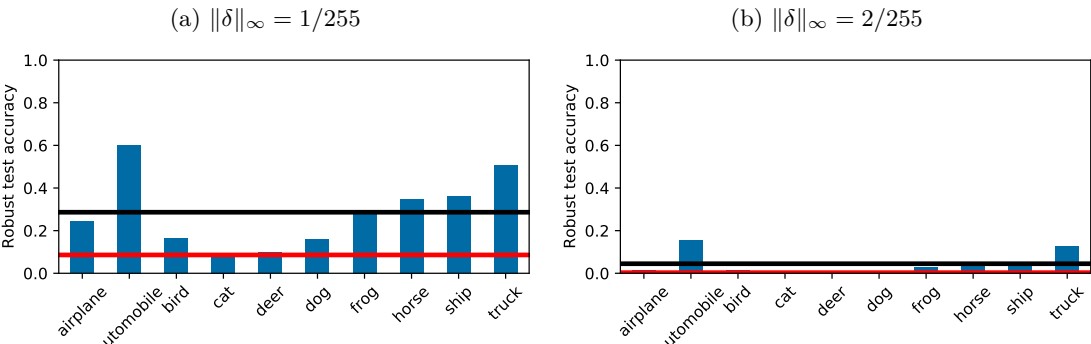

Figure 4: Robust test accuracy under different norm-ball sizes after *clean* training on CIFAR10. The non-uniform distribution over class accuracies, even after clean training, indicates that the inferior performance on some classes is not a consequence of adversarial training. Rather, the inhomogeneity after perturbation seems to be an inherent feature of the dataset.

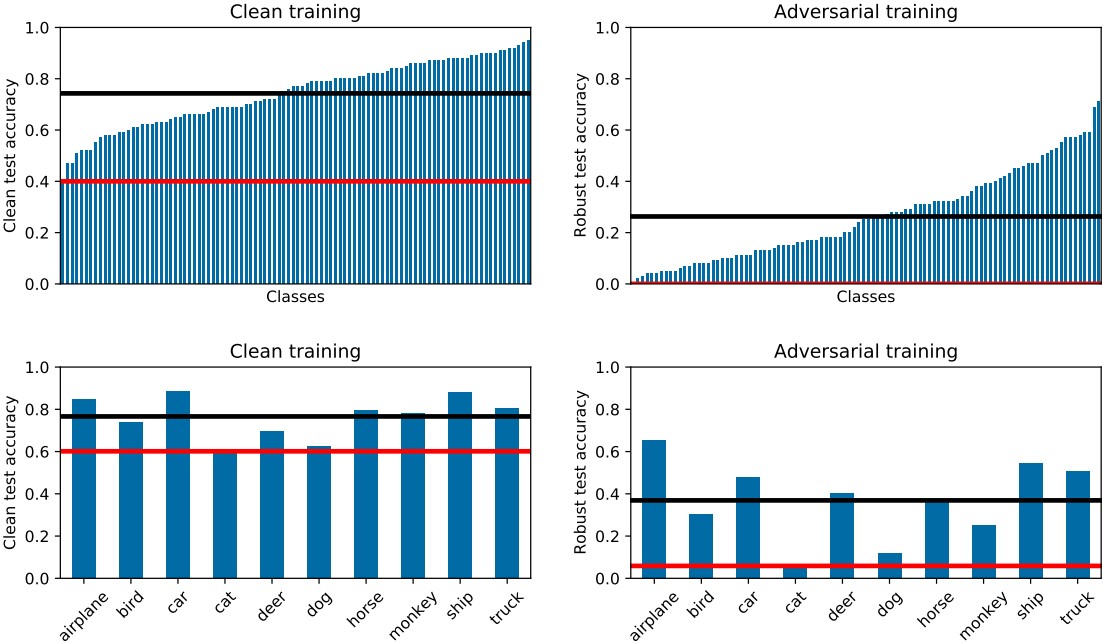

Figure 5: Clean training and adversarial training on CIFAR100 (top) and STL10 (bottom) using ERM-AT with PGD-7 attacks at train time and PGD-20 attacks at test time for the robust test accuracy. The CIFAR100 classes are sorted for convenience. Notice that CIFAR100 has a class with zero robust accuracy with adversarial training.

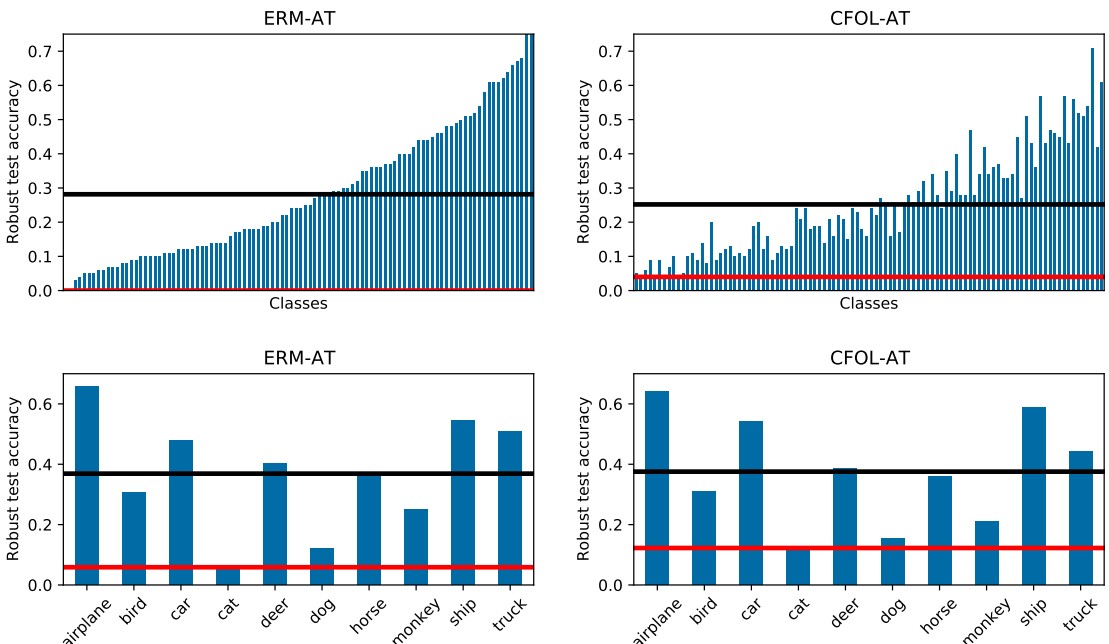

Figure 6: Robust class accuracy for CIFAR100 and STL10 respectively. Vertical black error bars indicate one standard deviation. The red and black horizontal line indicates the minimum and average respectively. The classes on both ERM-AT and CFOL-AT are ordered according to the accuracy on ERM-AT to make comparison easier.

Table 6: For fair comparison we also consider *early stopping based on the worst class accuracy* on the hold-out set. As can be observed the results for CFOL-AT do not differ significantly from early stopping using the average robust accuracy, so standard training setups do not have to be modified further.

| | | | ERM-AT | CFOL-AT | LCVaR-AT | FOL-AT |
|---|---|---|---|---|---|---|
| CIFAR10 | $acc_{clean}$ | Average | 0.8348 | 0.8336 | 0.8298 | **0.8370** |
| | | 20% tail | 0.6920 | **0.7460** | 0.6640 | 0.7045 |
| | | Worst class | 0.6710 | **0.7390** | 0.6310 | 0.7000 |
| | $acc_{rob}$ | Average | 0.4907 | 0.4939 | **0.5125** | 0.4982 |
| | | 20% tail | 0.2895 | **0.3335** | 0.2480 | 0.2840 |
| | | Worst class | 0.2860 | **0.3260** | 0.2450 | 0.2780 |
| CIFAR100 | $acc_{clean}$ | Average | **0.5824** | 0.5537 | 0.5800 | 0.5785 |
| | | 20% tail | 0.3115 | **0.3550** | 0.3065 | 0.3105 |
| | | Worst class | 0.1500 | **0.2200** | 0.1500 | 0.1700 |
| | $acc_{rob}$ | Average | 0.2622 | 0.2483 | **0.2629** | 0.2552 |
| | | 20% tail | 0.0540 | **0.0790** | 0.0605 | 0.0595 |
| | | Worst class | 0.0200 | **0.0400** | 0.0100 | 0.0200 |

Table 7: Model performance on CIFAR10 (left) and Imagenette (right) under AutoAttack (Croce & Hein, 2020). The models still uses 7 steps of PGD at training time with a $\ell_\infty$-constraint of 8/255. Only at test time is the attack exchanged with AutoAttack under the same constraint. CFOL-AT is robust to AutoAttack in the sense that the worst class performance is still improved. However, as expected, the performance is worse for both methods in comparison with 20-step PGD based attacks.

| | | ERM-AT | CFOL-AT | | | ERM-AT | CFOL-AT |
|---|---|---|---|---|---|---|---|
| | Average | 0.8244 | **0.8342** | | Average | 0.8638 | **0.8650** |
| $acc_{clean}$ | 20% tail | 0.6590 | **0.7510** | $acc_{clean}$ | 20% tail | 0.7687 | **0.7890** |
| | Worst class | 0.6330 | **0.7390** | | Worst class | 0.7150 | **0.7709** |
| | Average | **0.4635** | 0.4440 | | Average | **0.5911** | 0.5838 |
| $acc_{rob}$ | 20% tail | 0.1465 | **0.2215** | $acc_{rob}$ | 20% tail | 0.3576 | **0.4087** |
| | Worst class | 0.1260 | **0.2070** | | Worst class | 0.2254 | **0.3187** |

Table 8: Comparison between different sizes of $\ell_\infty$-ball attacks on CIFAR10. The same constraint is used at both training and test time. When the attack size is increased beyond the usual 8/255 constraint we still observe that CFOL-AT increases the robust accuracy for the weakest classes while taking a minor drop in the average robust accuracy. Interestingly, the gap between ERM-AT and CFOL-AT seems to enlarge. See Appendix C.2 for more detail on the attack hyperparameters.

| | | | ERM-AT | CFOL-AT |
|---|---|---|---|---|
| | | Average | 0.8244 | **0.8308** |
| | $acc_{clean}$ | 20% tail | 0.6590 | **0.7120** |
| | | Worst class | 0.6330 | **0.6830** |
| $\|\delta\|_\infty \leq 8/255$ | | Average | **0.5138** | 0.5014 |
| | $acc_{rob}$ | 20% tail | 0.2400 | **0.3300** |
| | | Worst class | 0.2350 | **0.3200** |
| | | Average | 0.7507 | **0.7594** |
| | $acc_{clean}$ | 20% tail | 0.4565 | **0.6060** |
| | | Worst class | 0.3850 | **0.5730** |
| $\|\delta\|_\infty \leq 12/255$ | | Average | **0.4054** | 0.3873 |
| | $acc_{rob}$ | 20% tail | 0.1180 | **0.2095** |
| | | Worst class | 0.0930 | **0.1700** |

Table 9: We test the reweighted variant of CFOL-AT (described in Section 4.1) on CIFAR10 and observe similar results as for CFOL-AT. The experimental setup is described in Section 5.

| | $acc_{clean}$ | | | $acc_{rob}$ | | |
|---|---|---|---|---|---|---|
| | Average | 20% tail | Worst class | Average | 20% tail | Worst class |
| CFOL-AT (reweighted) | 0.8286 | 0.7465 | 0.7370 | 0.4960 | 0.3290 | 0.3100 |

Table 10: Mean and standard deviation computed over 3 independent executions using different random seeds for both ERM-AT and CFOL-AT on CIFAR10.

| | | ERM-AT | CFOL-AT |
|---|---|---|---|
| $\text{acc}_{\text{clean}}$ | Average | **0.8324** $\pm$ 0.0069 | 0.8308 $\pm$ 0.0054 |
| | 20% tail | 0.6467 $\pm$ 0.0273 | **0.7192** $\pm$ 0.0436 |
| | Worst class | 0.5747 $\pm$ 0.0506 | **0.6843** $\pm$ 0.0121 |
| $\text{acc}_{\text{rob}}$ | Average | **0.5158** $\pm$ 0.0033 | 0.5012 $\pm$ 0.0008 |
| | 20% tail | 0.2540 $\pm$ 0.0428 | **0.3393** $\pm$ 0.0398 |
| | Worst class | 0.2067 $\pm$ 0.0254 | **0.3083** $\pm$ 0.0515 |

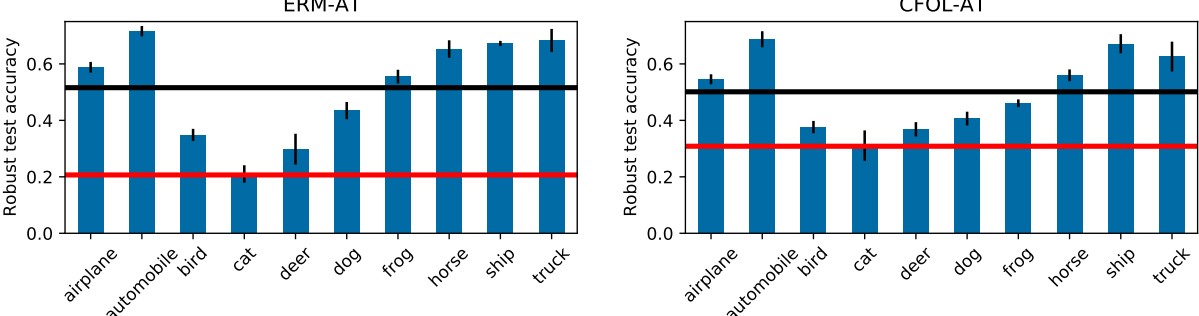

Table 11: A naive baseline improves the 20%-tail (marginally) over ERM-AT but is not able to improve the worst class. The baseline first runs ERM-AT to obtain the robust class accuracies on the test set. The negative accurcies are pushed through a softmax to obtain a distribution $q$ and ERM-AT is rerun with classes sampled from $p = \gamma \frac{1}{k} + (1 - \gamma)q$.

| | | $\gamma = 0.5$ | $\gamma = 0.9$ |
|---|---|---|---|
| $\text{acc}_{\text{clean}}$ | Average | **0.8384** | 0.8277 |
| | 20% tail | **0.6810** | 0.6470 |
| | Worst class | 0.6340 | **0.6430** |
| $\text{acc}_{\text{rob}}$ | Average | 0.5070 | **0.5094** |
| | 20% tail | **0.2760** | 0.2530 |
| | Worst class | 0.2180 | **0.2330** |

Table 12: We investigate the effect of using a diffent architecture (VGG-16 (Simonyan & Zisserman, 2014)) on CIFAR10. The worst classes for CFOL-AT remains improved over ERM-AT. Unsurprisingly the older network performs worse for both methods when compared with their ResNet-18 counterpart.

| | | ERM-AT | CFOL-AT |
|---|---|---|---|
| $\text{acc}_{\text{clean}}$ | Average | 0.7805 | **0.7904** |
| | 20% tail | 0.5385 | **0.6550** |
| | Worst class | 0.5020 | **0.6220** |
| $\text{acc}_{\text{rob}}$ | Average | **0.4799** | 0.4606 |
| | 20% tail | 0.2165 | **0.3075** |
| | Worst class | 0.1940 | **0.3070** |

Table 13: Experiments on Tiny ImageNet (Russakovsky et al., 2015) which consists of 100,000 images across 200 classes. CFOL-AT improves the average accuracy for the 123 worst classes without optimizing the hyperparameters (the hyperparameters remains the same as for CIFAR10, CIFAR100 and STL10).

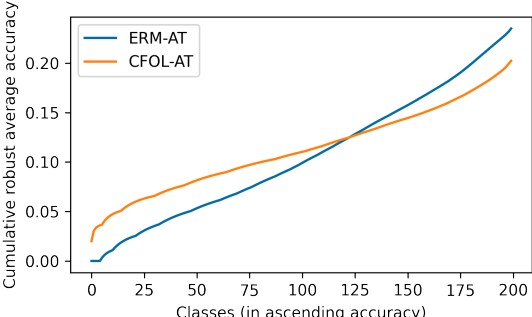

|  |  | ERM-AT | CFOL-AT |
|---|---|---|---|
| $\mathrm{acc_{clean}}$ | Average | **0.4784** | 0.4606 |
|  | 20% tail | 0.1725 | **0.2685** |
|  | Worst class | 0.0200 | **0.1600** |
| $\mathrm{acc_{rob}}$ | Average | **0.2349** | 0.2023 |
|  | 20% tail | 0.0445 | **0.0735** |
|  | Worst class | 0.0000 | **0.0200** |

Table 14: Imbalanced CIFAR10 with imbalance factor 10 such that the majority class and the minority class has 5000 and 500 training samples respectively. We early stop based on the uniformly distributed test set. Three variants of CFOL are considered. Both the mixing distribution and initialization of the adversary $q^0$ can be either uniform over classes (U) or according to the empirical training distribution (E). For instance if $q^0$ follows the empirical distribution (E) and the mixing distribution is uniform (U), then we suffix CFOL-AT with "EU". We observe that CFOL-AT EE in particular improves the worst class accuracies, while the average accuracy incurs an even smaller drop than under uniform CIFAR10 (Table 1). It is interesting to understand the seeming tradeoff between clean accuracy, average robust accuracy and robust worst class accuracy cause by different instantiations of CFOL.

|  |  | ERM-AT | CFOL-AT UU | CFOL-AT EU | CFOL-AT EE |
|---|---|---|---|---|---|
| $\mathrm{acc_{clean}}$ | Average | 0.7300 | **0.7772** | 0.7708 | 0.7586 |
|  | 20% tail | 0.5770 | **0.6780** | 0.6450 | 0.6420 |
|  | Worst class | 0.5410 | **0.6730** | 0.6360 | 0.6000 |
| $\mathrm{acc_{rob}}$ | Average | 0.4065 | 0.3988 | **0.4098** | 0.3991 |
|  | 20% tail | 0.2200 | 0.2550 | **0.2625** | **0.2625** |
|  | Worst class | 0.2140 | 0.2220 | 0.2240 | **0.2590** |

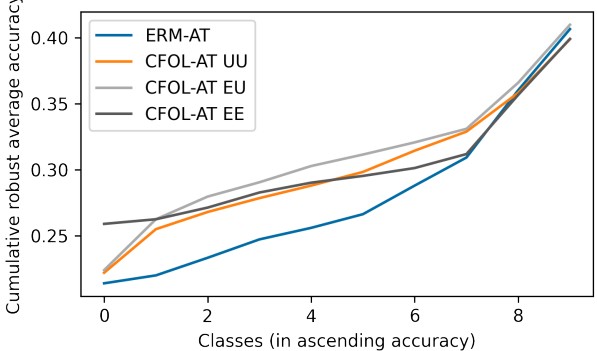

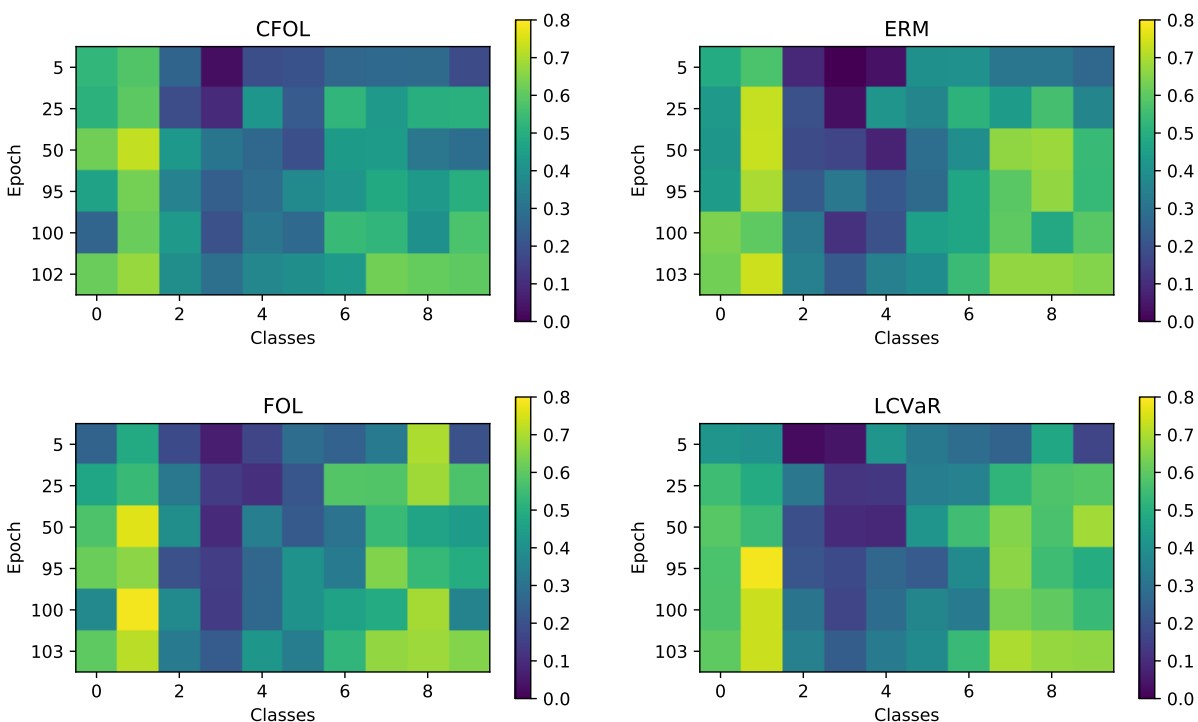

Figure 7: The evolution of the robust validation accuracies across epochs. The final epoch is when the model was early stopped based on the average robust accuracy. There is a clear pattern in what classes are harder (class 2-5). Both CFOL and FOL have a curious non-monotonic evolution for the class conditional accuracy. Notice for instance that both methods drop in accuracy for the 0th class at the stepsize change at epoch 100.

Table 15: Imagenette classification using ResNet-18.

|  |  | **ERM-AT** | **CFOL-AT** |
|---|---|---|---|
| $\mathrm{acc_{clean}}$ | Average | 0.8638 | **0.8650** |
|  | 20% tail | 0.7687 | **0.7890** |
|  | Worst class | 0.7150 | **0.7709** |
| $\mathrm{acc_{rob}}$ | Average | **0.6285** | 0.6181 |
|  | 20% tail | 0.4026 | **0.4534** |
|  | Worst class | 0.2850 | **0.3912** |

## C.1 Hyperparameters

In this section we provide additional details to the hyperparameters specified in Section 5.

For CFOL-AT we use the same parameters as for ERM-AT described in Section 5. We set $\gamma = \frac{1}{2}$ and the adversarial step-size $\eta = 2 \cdot 10^{-6}$ across all experiments, if not otherwise noted. One exception is Imagenette were number of iterations are roughly 5 times fewer due to the smaller dataset. Thus, $\eta$ is picked 2.5 times larger as suggested by theory through $\eta = \tilde{\mathcal{O}}(1/\sqrt{T})$ in Theorem 1. CFOL-AT seems to be reasonable robust to step-size choice as seen in Table 16. Similarly for FOL-AT we use the same parameters as for ERM-AT and set $\eta = 1 \cdot 10^{-7}$ after optimizing based on the worst class robust accuracy on CIFAR10.

The adversarial step-size picked for both FOL-AT and CFOL-AT is smaller in practice than theory suggests. This suggests that the mistake bound in Assumption 1 is not satisfied for any sequence. Instead we rely on

Table 16: Hyperparameter exploration for CFOL-AT, LCVaR-AT and FOL-AT on CIFAR10.

| | Parameters | | $\text{acc}_{\text{rob}}$ | | $\text{acc}_{\text{clean}}$ | |
| | $\eta$ | $\alpha$ | Average | Worst class | Average | Worst class |
|---|---|---|---|---|---|---|
| CFOL-AT | $1 \times 10^{-6}$ | - | 0.5076 | 0.2700 | 0.8372 | 0.7110 |
| | $2 \times 10^{-6}$ | - | 0.5014 | 0.3200 | 0.8308 | 0.6830 |
| | $5 \times 10^{-6}$ | - | 0.4963 | 0.3000 | 0.8342 | 0.7390 |
| | $1 \times 10^{-5}$ | - | 0.4927 | 0.2850 | 0.8314 | 0.7160 |
| LCVaR-AT | - | 0.1 | 0.4878 | 0.1590 | 0.7909 | 0.5120 |
| | - | 0.2 | 0.5116 | 0.1250 | 0.8274 | 0.4710 |
| | - | 0.5 | 0.5104 | 0.1890 | 0.8177 | 0.5550 |
| | - | 0.8 | 0.5169 | 0.2440 | 0.8259 | 0.6340 |
| | - | 0.9 | 0.5249 | 0.1760 | 0.8233 | 0.5170 |
| FOL-AT | $1 \times 10^{-7}$ | - | 0.5106 | 0.2370 | 0.8280 | 0.6210 |
| | $5 \times 10^{-7}$ | - | 0.5038 | 0.2180 | 0.8372 | 0.6100 |
| | $1 \times 10^{-6}$ | - | 0.4885 | 0.2190 | 0.8237 | 0.5880 |
| | $5 \times 10^{-6}$ | - | 0.4283 | 0.1560 | 0.8043 | 0.5380 |

the sampling process to be only mildly adversarial initially as implicitly enforced by the small adversarial step-size. It is an interesting future direction to incorporate this implicit tempering directly into the mixing parameter $\gamma$ instead.

For LCVaR-AT we optimized over the size of the uncertainty set by adjusting the parameter $\alpha$ on CIFAR10. This leads to the choice of $\alpha = 0.8$ which we use across all subsequent datasets. The hyperparameter exploration can be found in Table 16.

## C.2 Experimental setup

In this section we provide additional details for the experimental setup specified in Section 5. We use one GPU on an internal cluster. The experiments are conducted on the following five datasets:

**CIFAR10** includes 50,000 training examples of $32 \times 32$ dimensional images and 10 classes.

**CIFAR100** includes 50,000 training examples of $32 \times 32$ dimensional images and 100 classes.

**STL10** includes 5000 training examples of $96 \times 96$ dimensional images and 10 classes.

**TinyImageNet** 100,000 training examples of $64 \times 64$ dimensional images and 200 classes.

**Imagenette** [1] 9469 training examples of $160 \times 160$ dimensional images and 10 classes.

For all experiment we use data augmentation in the form of random cropping, random horizontal flip, color jitter and 2 degrees random rotations. Prior to cropping a black padding is added with the exception of Imagenette. For Imagenette we follow the standard for ImageNet evaluation and center crop the validation and test images.

**Early stopping** As noted in Section 5 we use the *average* robust accuracy to early stop the model. In contrast with common practice though, we use a validation set instead of the test set to avoid overfitting to the test set. The class accuracies across the remaining two datasets, CIFAR100 and STL10, can be found in Figure 6. We also include results when the model is early stopped based on the worst class accuracy on the validation set in Table 6.

---

[1]`https://github.com/fastai/imagenette`

**Attack parameters** A radius of $^8/_{255}$ is used for the $\ell_\infty$-constraint attack unless otherwise noted. For training we use 7 steps of PGD and a step-size of $^2/_{255}$. At test time we use 20 steps of PGD with a step-size of $2.5 \times \frac{8/255}{20}$. For $^{12}/_{255}$-bounded attacks in Table 8 we scale the training step-size and test step-size proportionally.

**Larger models** For the larger model experiments in Table 4 we train a Wide ResNet-34-10 using CFOL-AT (with $\gamma = 0.8$). Similarly to the pretrained weights from the literature, we early stop based on the test-set. We compare against TRADES (Zhang et al., 2019) (Wide ResNet-34-10) and the training setup of Madry et al. (2017), which we throughout have denoted as ERM-AT, using shared weights from Engstrom et al. (2018) (ResNet-50).

**Mean & standard deviation** In Table 10 we apply CFOL-AT and ERM-AT to CIFAR10 for multiple different random seeds and verify that the observations made in Section 5 remains unchanged.

### C.3 Implementation

We provide pytorch pseudo code for how CFOL can be integrated into existing training setups in Listing 1. FOL is similar in structure, but additionally requires associating a unique index with each training example. This allows the sampling method to re-weight each example individually.

For LCVaR we use the implementation of (Xu et al., 2020), which uses the dual CVaR formulation described in Appendix B. More specifically, LCVaR uses the variant which finds a closed form solution for $\lambda$, since this was observed to be both faster and more stable in their work.

Listing 1: Pseudo code for CFOL.

```python
from torch.utils.data import DataLoader
from class_sampler import ClassSampler

sampler = ClassSampler(dataset, gamma=0.5)
dataloader = DataLoader(dataset, ..., sampler=sampler)

...

# Training loop:
for img,y in iter(dataloader):

  # attack img
  # compute gradients and update model
  # compute logits

  adv_loss = logits.argmax(dim=-1) != y
  sampler.batch_update(y, eta * adv_loss)
```

