# OpenReview forum: "Revisiting adversarial training for the worst-performing class"
_TMLR — Accepted by TMLR_

### Review · Reviewer_oVMu · 2022-10-09

**Summary Of Contributions:**

This paper studies the problem of adversarial training where the desired metric is to ensure the robustness of the worst-case class as opposed to the average-case class. In contrast to standard training, in adversarial training there is an even larger gap between the best and the worst performing classes (with respect to robust accuracy).

+ The authors propose an online-learning type algorithm that adjust the adversarial training algorithm to optimize for the worst-case class.
+ The authors prove a convergence rate for the algorithm
+ The authors evaluate their approach on CIFAR10, CIFAR100, and STL10 and compare to various other adversarial training and robust optimization approaches.

**Requested Changes:**

I would suggest the authors run a similar comparison on ImageNet (or a restricted/smaller version thereof). Other different settings could also work as well---any additional setting would certainly help argue for a broader argument that the approach works on more than just classification of 32x32 images.

While I think this would be nice to have, the paper includes other non-empirical results such as convergence guarantees, and thus this would simply strengthen the work in my view and is not critical.

**Strengths And Weaknesses:**

The main strengths of the submission are:
+ The approach appears to be theoretically grounded, with a convergence rate guarantee. In practice, this allows the algorithm to only need to optimize over the worst class at each iteration instead of all classes at once. Though I am less of an expert on the online learning literature and cannot fully vouch for its technical correctness.
+ Experimentally, the results consistently lead to sizeable improvements for the worst-case class. The comparison points in the adversarial robustness space are reasonable (standard adversarial training & trades), which understandably perform worse as they do not optimize for the worst case class.
+ The method has only a small amount of overhead,

One weaker aspect of this paper is that the experimental results are on 3 fairly similar datasets (CIFAR10/CIFAR100/STL10). It would be nice to see results on substantially different datasets to demonstrate the broader applicability of this method, i.e. on Imagenet or other non-32x32 pixel classification tasks.

---

> ### Author Response · Authors · 2022-10-19
> **Response to Reviewer oVMu**
>
>
> We thank the reviewer for the positive comments and address the suggestion below.
>
> >  I would suggest the authors run a similar comparison on ImageNet (or a restricted/smaller version thereof).
>
> We have carried out additional experiments on Imagenette which consists of 10 classes from Imagenet (even this restricted subset took close to 30 hours per run). The results can be found in the table below and in the updated manuscript (Table 15), where CFOL still improves the 20% worst tail while suffering a small reduction in the average accuracy.
>
> We would also like to draw to the attention that the paper already contained experiments on 96x96 images (STL10) and 64x64 images (TinyImageNet), please see the Table 2 and 13 as well as Section C.2 for details.
>
> |                                                        |             | **ERM-AT**    | **CFOL-AT**   |
> |--------------------------------------------------------|-------------|--------------------|--------------------|
> |  | Average     | 0.8638             | **0.8650** |
> | $\mathrm{{acc}}_{{\mathrm{{clean}}}}$                                                       | 20% tail   | 0.7687             | **0.7890** |
> |                                                        | Worst class | 0.7150             | **0.7709** |
> |    | Average     | **0.6285** | 0.6181             |
> | $\mathrm{{acc}}_{{\mathrm{{rob}}}}$                                                       | 20% tail   | 0.4026             | **0.4534** |
> |                                                        | Worst class | 0.2850             | **0.3912** |

---

### Review · Reviewer_ujcb · 2022-10-26

**Summary Of Contributions:**

The paper considers a class-conditional form of adversarial training where an adversary is further allowed to choose the worst-performing class $y$ in the minimax optimization - which is related to some safety critical applications. The paper relaxes the given formulation in discrete optimization of $y$ with a continuous online learning algorithm and proposes techniques to improve convergence rate and unbiasedness of the algorithm, as well as performing a convergence rate analysis. Experimental results on CIFAR-10/100 and STL show that the proposed method successfully recovers a proper distribution of $y$ suitable for the worst-class adversarial training, improving the worst-class robust accuracy of adversarial training compared to baselines.

**Broader Impact Concerns:**

None that I am aware of.

**Requested Changes:**

See the Weakness above.

**Strengths And Weaknesses:**

**Strength**

* The paper is clearly written and mathematically sound.
* The claims are backed up both in theoretically and empirically
* The experimental results consistently demonstrate the effectiveness over the baselines


**Weakness**

* The empirical significance of the proposed method could be questionable, especially in CIFAR-100 which has more number of classes and thus more realistic, e.g., there is only 0.00 → 0.04 of accuracy gain in worst-case robustness
* The paper specifies that the major evaluation metric (those reported in the main tables) is from PGD-20, although it also reports AutoAttack results in Appendix. I think the stronger attack of AutoAttack should be the major evaluation protocol to report, given that the paper proposes a modification on adversarial training that is sometimes witnessed to show a false defense claim without a thorough evaluations.
* Any discussion on the (practical) training overhead compared to standard AT would help for practitioners to use the proposed method.

---

> ### Author Response · Authors · 2022-10-27
> **Response to Reviewer ujcb**
>
> We thank the reviewer for the feedback and address the concerns below.
>
> - **Practical computational overhead** We appreciate the remark for which we list the total runtime for the experiments in the table below. As seen, CFOL-AT increases the runtime with roughly 1%-5%. More classes (CIFAR100) leads to a slightly larger overhead, while larger images (STL10) makes the relative cost of the adversarial distribution update smaller. Since the overhead is only marginal we have not done anything to optimize the adversarial update rule (the batch is currently processed with a for-loop). If a smaller compute overhead is desired, this computation could be parallelized.
>
> |          | ERM        | CFOL       | Increase |
> |----------|------------|------------|----------|
> | STL10    | 2h 32m 53s | 2h 34m 53s | ~1.3%    |
> | CIFAR100 | 5h 51m 0s  | 6h 7m 11s  | ~4.5%    |
> | CIFAR10  | 5h 44m 46s | 5h 53m 46s | ~2.3%    |
>
> - **Concerning attack evaluation** It might be worth clarifying that we are not proposing a different attack on the image level. CFOL-AT is in fact still using the PGD attack, which does not rely on spurious mechanism such as gradient masking [1] to fake robustness. The comparison provided under PGD-20 between the four considered methods remains fair, since they all use the same underlying PGD attack. To be extra cautious we also included AutoAttack for CIFAR10 in the appendix (as the reviewer pointed out) where the conclusions remain the same. The AutoAttack on CIFAR10 took 2 hours per run for the test time evaluation alone, so given it did not change the core insight, we refrained from using AutoAttack throughout. However, we have additionally evaluated Imagenette with AutoAttack (taking ~11 hours per evaluation). The results can be found below and in table 7 of the updated manuscript, where we can confirm that the conclusions remain the same.
>
> |   |             | **ERM-AT**    | **CFOL-AT**   |
> |--------------------------------------------------------|-------------|--------------------|--------------------|
> |   | Average     | 0.8638             | **0.8650** |
> | $\mathrm{{acc}}_{{\mathrm{{clean}}}}$ | 20% tail   | 0.7687             | **0.7890** |
> |   | Worst class | 0.7150             | **0.7709** |
> |   | Average     | **0.5911** | 0.5838             |
> | $\mathrm{{acc}}_{{\mathrm{{rob}}}}$   | 20% tail   | 0.3576             | **0.4087** |
> |   | Worst class | 0.2254             | **0.3187** |
>
> - **Challenge of more classes** We agree with the reviewer that this is seemingly a non-impressive difference. However, please let us emphasize that this is the worst perturbations of the input image. To provide some perspective, we notice that the other methods accepted for their worst class performance (i.e. FOL, LCVaR), score also 0% in the same benchmark.
>
>     Therefore, we would like to point out the hardness of the problem altogether, which makes it an important problem to study.
>
> [1] https://arxiv.org/pdf/1602.02697.pdf

---

### Review · Reviewer_Tj9o · 2022-11-06

**Summary Of Contributions:**

The setting:
The setting is robust learning to perturbations at test time (adversarial examples), where instead of minimizing the average loss, the goal is to minimize the loss of the class with the lowest accuracy.
So instead of a min-max problem objective, the resulting objective consists of another maximization component over the classes, and is named "worst class-conditioned risk".

Contributions:
The authors suggest a method from online learning - exponential weight in the bandit setting (that is - Exp3).

Moreover, they provide a convergence guarantee with high probability.

Finally, the effectiveness of the suggested method is tested empirically, on CIFAR100 and STL10.



**Broader Impact Concerns:**

-

**Requested Changes:**

No request for changes.



**Strengths And Weaknesses:**

-The model is well-motivated, in some applications, such an imbalance could be problematic.
This is also supported by an example that shows a larger variance in losses between classes than in the standard setting.
(The perspective of adversarial examples where the adversary chooses what class to evaluate on in addition
to the perturbation, seems unrealistic to me. However, I believe that problem still models some interesting scenarios).

-The suggested method is intuitive and seems effective in rebalancing the loss over different classes.

-The technical novelty of the convergence guarantee is only marginally novel and follows mainly from known analysis (if I'm not missing something), but it is good to include it.

---

> ### Author Response · Authors · 2022-11-18
> **Response to Reviewer Tj9o**
>
> We thank the reviewer for the positive feedback.
>
> > The perspective of adversarial examples where the adversary chooses what class to evaluate on in addition to the perturbation, seems unrealistic to me. However, I believe that problem still models some interesting scenarios
>
> We agree with the reviewer and would like to clarify. We *model* the problem as an adversarial threat model in which the attacker chooses what class to evaluate on in addition to the perturbation. From this formulation we develop CFOL-AT, which ultimately provides some level of robustness across *all* classes. As the reviewer points out this property can be interesting even if there is no malicious adversary picking the worst class. We have updated the wording in the introduction and the conclusion which hopefully clarifies.

---

### Author Response · Authors · 2022-12-07
**A kind reminder**

Dear reviewers,

As the discussion period is coming to an end we would like to thank you for your valuable feedback. We would greatly appreciate that you let us know if our changes and additional experiments have addressed your concerns. We remain available to answer any further questions.

---

### Decision · Action_Editors · 2022-12-26

**Recommendation:** Accept as is

**Comment:**

All reviewers were generally satisfied with the authors' response to their review and recommended acceptance of the paper.

**Audience:**

The paper should be interesting for researcher working on adversarial robustness, as well as researchers interested in distribution-shift robustness more broadly, since worst-case class performance can be interpreted as robustness to label shift.

**Claims And Evidence:**

The main claims of the papers is that adversarial robustness drops significantly on the worst-performing class, and that algorithms directly target worst-class adversarial robustness can help mitigate this phenomenon. Both claims are well supported by experiments, and the proposed algorithm is also justified via theoretical analysis.